# A time-resolved picture of our Milky Way's early formation history

Maosheng Xiang[1 ✉] & Hans-Walter Rix[1 ✉]

The formation of our Milky Way can be split up qualitatively into different phases that resulted in its structurally different stellar populations: the halo and the disk components[1–3]. Revealing a quantitative overall picture of our Galaxy's assembly requires a large sample of stars with very precise ages. Here we report an analysis of such a sample using subgiant stars. We find that the stellar age–metallicity distribution $p(\tau, \text{[Fe/H]})$ splits into two almost disjoint parts, separated at age $\tau \simeq 8$ Gyr. The younger part reflects a late phase of dynamically quiescent Galactic disk formation with manifest evidence for stellar radial orbit migration[4–6]; the other part reflects the earlier phase, when the stellar halo[7] and the old $\alpha$-process-enhanced (thick) disk[8,9] formed. Our results indicate that the formation of the Galaxy's old (thick) disk started approximately 13 Gyr ago, only 0.8 Gyr after the Big Bang, and 2 Gyr earlier than the final assembly of the inner Galactic halo. Most of these stars formed around 11 Gyr ago, when the Gaia-Sausage-Enceladus satellite merged with our Galaxy[10,11]. Over the next 5–6 Gyr, the Galaxy experienced continuous chemical element enrichment, ultimately by a factor of 10, while the star-forming gas managed to stay well mixed.

To unravel the assembly history of our Galaxy we need to learn how many stars were born when, from what material and on what orbits. This requires precise age determinations for a large sample of stars that extend to the oldest possible ages (around 14 Gyr)[9,12]. Subgiant stars, which are stars sustained by hydrogen shell fusion, can be unique tracers for such purposes, as they exist in the brief stellar evolutionary phase that permits the most precise and direct age determination, because their luminosity is a direct measure of their age. Moreover, the chemical element compositions determined from the spectra of their photosphere surfaces accurately reflect their birth material composition billions of years ago. This makes subgiants the best practical tracers of Galactic archaeology, even compared to main-sequence turn-off stars, whose surface abundances may be altered by atomic diffusion effects[13]. However, because of the short lifetime of their evolutionary phase, subgiant stars are relatively rare, and large surveys are essential to build a large sample of these objects with good spectra, which have not been available in the past.

With the recent data release (eDR3) of the Gaia mission[14,15] and the recent data release (DR7) of the LAMOST spectroscopic survey[16,17], we identify a set of approximately 250,000 subgiant stars based on their position in the effective temperatures ($T_{\text{eff}}$)–absolute magnitude ($M_K$) diagram (Fig. 1a). The ages ($\tau$) of these subgiant stars are estimated by fitting to the Yonsei–Yale (YY) stellar isochrones[18] with a Bayesian approach, which draws on the astrometric distances (parallaxes), apparent magnitudes (fluxes), spectroscopic chemical abundances ([Fe/H], [$\alpha$/Fe] where $\alpha$ refers to $\alpha$ elements Mg, Si, Ca, Ti), $T_{\text{eff}}$ and $M_K$. As summarized in Fig. 1b, the sample stars have a median relative age uncertainty of only 7.5% across the age range from 1.5 Gyr to the age of the Universe (13.8 Gyr; ref.[19]). The lower age limit of our sample is inherent to our approach: younger and hence more luminous subgiants can be confused with a different stellar evolutionary phase, the horizontal

branch phase for far older stars, which would cause serious sample contamination. This sample constitutes a 100-fold leap in sample size for stars with comparably precise and consistent age estimates[20,21]. In addition, it is a large sample that covers a large spatial volume across the Milky Way (Fig. 1c) and most of the pertinent range in age and in metallicity (1.5 Gyr < $\tau$ < 13.8 Gyr, and −2.5 < [Fe/H] < 0.4). The sample also has a straightforward spatial selection function that allows us to estimate the space density of the tracers. These ingredients enable an alternative view of the Milky Way's assembly history, especially the early formation history.

## Our Galaxy's stellar age–metallicity distribution

The photospheric metallicity of any subgiant star of age $\tau$ reflects the element composition of the gas from which it formed at the epoch $\tau$ Gyr ago. The overall distribution of these stellar metallicities at different epochs, $p(\tau, \text{[Fe/H]})$, thus encodes the chemical enrichment history of our Milky Way galaxy. Figure 2a presents this distribution for our data. It shows that the age–metallicity distribution exhibits a number of prominent and distinct sequences, including at least two age-separated sequences with [Fe/H] > −1, and a sequence of exclusively old stars at low metallicity, [Fe/H] < −1. The density of $p(\tau, \text{[Fe/H]})$ may change with stellar orbit or Galactocentric radius, in the range our sample covers (6–14 kpc; Fig. 1). Yet, the 'morphology' of the distribution varies only slightly, enabling us to focus on the radially averaged distribution $p(\tau, \text{[Fe/H]})$ here.

It turns out that the complexity of $p(\tau, \text{[Fe/H]})$ (Fig. 2a) can be unravelled by dividing the sample into two subsamples using stellar quantities that are neither $\tau$ nor [Fe/H]: the angular momentum $J_\phi$ (also denoted as $L_z$) and the '$\alpha$-enhancement', [$\alpha$/Fe]. Extensive observations indicate that the majority of stars in the Milky Way formed from gradually enriched gas on high-angular momentum orbits, or the extended

[1]Max-Planck Institute for Astronomy, Koenigstuhl 17, Heidelberg, Germany. ✉e-mail: mxiang@mpia.de; rix@mpia.de

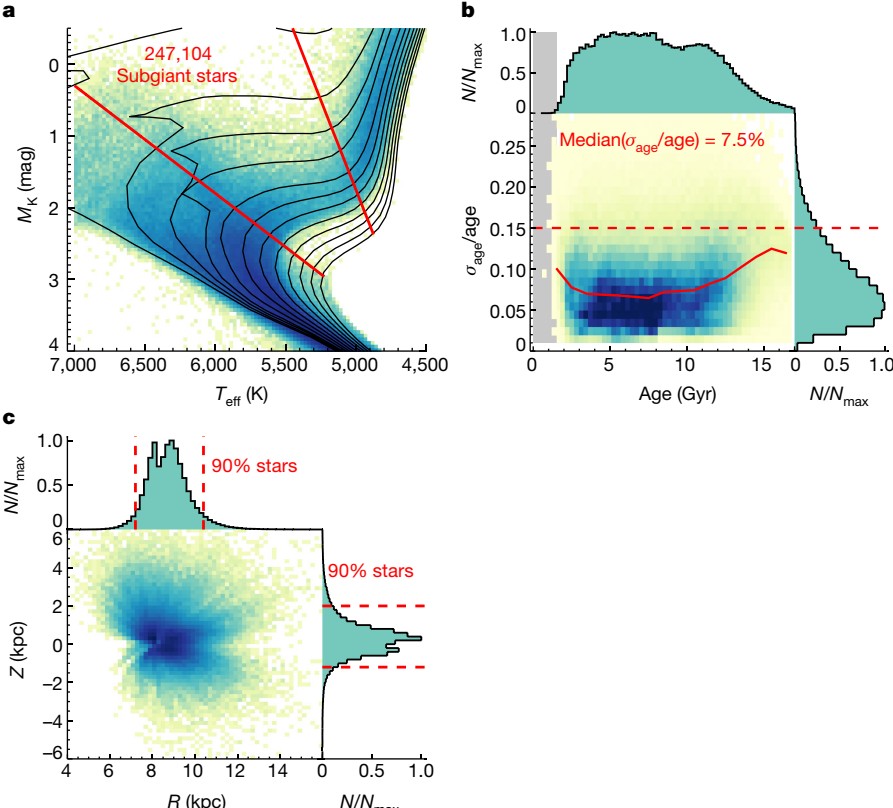

**Fig. 1 | The subgiant star sample with precise ages. a**, Illustration of the subgiant selection in the $T_{eff}$–$M_K$ diagram, shown for the solar metallicity bin of $-0.1 <$ [Fe/H] $< 0.1$. In total, the subgiant sample contains 247,104 stars. The solid curves are isochrones from the YY stellar evolution models[18] for solar metallicity ([Fe/H] $= 0$, [$\alpha$/Fe] $= 0$) for ages of 1, 2, 3, 4, 6, 8, 10, 12, 14, 16, 18 and 20 Gyr, illustrating how stellar ages can be determined from the position in the $T_{eff}$–$M_K$ diagram if [Fe/H] is known. The two straight lines bracket the region within which we define our subgiant star sample. **b**, Distribution in the relative age precision as a function of age: the mode of this precision distribution is at 6% and the median at 7.5%. For the subsequent analysis we will only use stars with a relative age precision of less than 15% (horizontal dashed line). Histograms in the top and right are normalized to the peak value $N_{max}$. **c**, Spatial distribution of our subgiant sample stars in the $R$–$Z$ plane of Galactic cylindrical coordinates. The full extent of the Galactocentric radius in the sample is 6 kpc $\lesssim R \lesssim$ 14 kpc and that of the distance from the Galactic mid-plane is $-5$ kpc $\lesssim Z \lesssim$ 6 kpc. The bulk of the sample (90%) covers 7.2 kpc $\lesssim R \lesssim$ 10.4 kpc and $-1.2$ kpc $\lesssim Z \lesssim$ 2 kpc, as illustrated by the dashed lines.

('thin') disk[4,22], at high $J_\phi$ and low [$\alpha$/Fe]. It is also well established that the distribution of Galactic stars in the [$\alpha$/Fe]–[Fe/H] plane is bimodal, with a high-$\alpha$ sequence reflecting rapid enrichment and a low-$\alpha$ sequence reflecting gradual enrichment, which indicates a natural way to divide any sample in the [$\alpha$/Fe]–[Fe/H] plane[8]. This inspired our approach to divide our sample into two, separating the dominant sample portion of gradually enriched disk stars with high angular momentum from the rest. Specifically, we used the cut

$$
\begin{cases}
J_\phi > 1500 \text{ kpc . km/s} \text{ and} \\
\begin{cases}
[\alpha/Fe] > 0.16, & \text{if } [\text{Fe/H}] > -0.5, \\
[\alpha/Fe] < -0.16[\text{Fe/H}] + 0.08, & \text{if } [\text{Fe/H}] > -0.5,
\end{cases}
\end{cases}
\tag{1}
$$

which is illustrated as a yellow shaded area in Fig. 2b, c. The resulting subsamples in the $\tau$–[Fe/H] plane are shown in Fig. 2d, e, where it is crucial to recall that the sample split involved neither of the quantities on the two axes, $\tau$ and [Fe/H]. As we want to focus first on the Milky Way's elemental enrichment history, rather than its star-formation history, we normalize the distribution $p(\tau, [\text{Fe/H}])$ at each [Fe/H] to yield $p(\tau | [\text{Fe/H}])$, the age distribution at a given [Fe/H].

Figure 2d, e shows that this cut in angular momentum and [$\alpha$/Fe] separates the Milky Way's enrichment history neatly into two distinct age regimes, with a rather sharp transition at $\tau \simeq 8$ Gyr. We will therefore refer to these two portions, not clearly apparent in earlier data, as

$p(\tau | [\text{Fe/H}])_{\text{late}}$ and $p(\tau | [\text{Fe/H}])_{\text{early}}$. The distribution of $p(\tau | [\text{Fe/H}])_{\text{late}}$ clearly exhibits a V-shape[23]. This shape is presumably a consequence of the secular evolution of the dynamically quiescent disk; the metal-rich ([Fe/H] $\gtrsim -0.1$) branch arises from stars that have migrated from the inner disk to near the Solar radius. The slope of that branch in $p(\tau | [\text{Fe/H}])_{\text{late}}$ then results from the (negative) radial metallicity gradient in the disk[1] and the fact that the stars that have migrated more needed more time to do so, and are hence older. Analogously, we presume the lower branch of $p(\tau | [\text{Fe/H}])_{\text{late}}$ at [Fe/H] $\lesssim -0.1$ to arise from stars that were born further out and have migrated inwards[6]. A quantitative comparison with secular evolution models of the Galactic disk[4,22] is part of separate ongoing work.

The older stars, reflected in $p(\tau | [\text{Fe/H}])_{\text{early}}$, show two prominent sequences with distinct [Fe/H]($\tau$) relations. The stars with $-2.5 <$ [Fe/H] $< -1.0$ reflect the well-established stellar halo population of our Milky Way, whereas the more metal-rich sequence ([Fe/H] $\gtrsim -1$) reflects the Milky Way's inner, high-$\alpha$ (thick) disk[24]; this designation as an old disk component is also justified by the stars' angular momentum, as we will show below.

The morphology of the old disk sequence in $p(\tau | [\text{Fe/H}])_{\text{early}}$ is the most striking feature in Fig. 2e; it reveals an exceptionally clear, continuous and tight age–metallicity relation from [Fe/H] $\lesssim -1$ at 13 Gyr ago all the way to [Fe/H] $\simeq 0.5$ at 7 Gyr ago. A simple model for $p(\tau | [\text{Fe/H}])$ of this sequence (Supplementary Information) finds an intrinsic age dispersion of less than 0.82 Gyr at a given [Fe/H] across this 6 Gyr

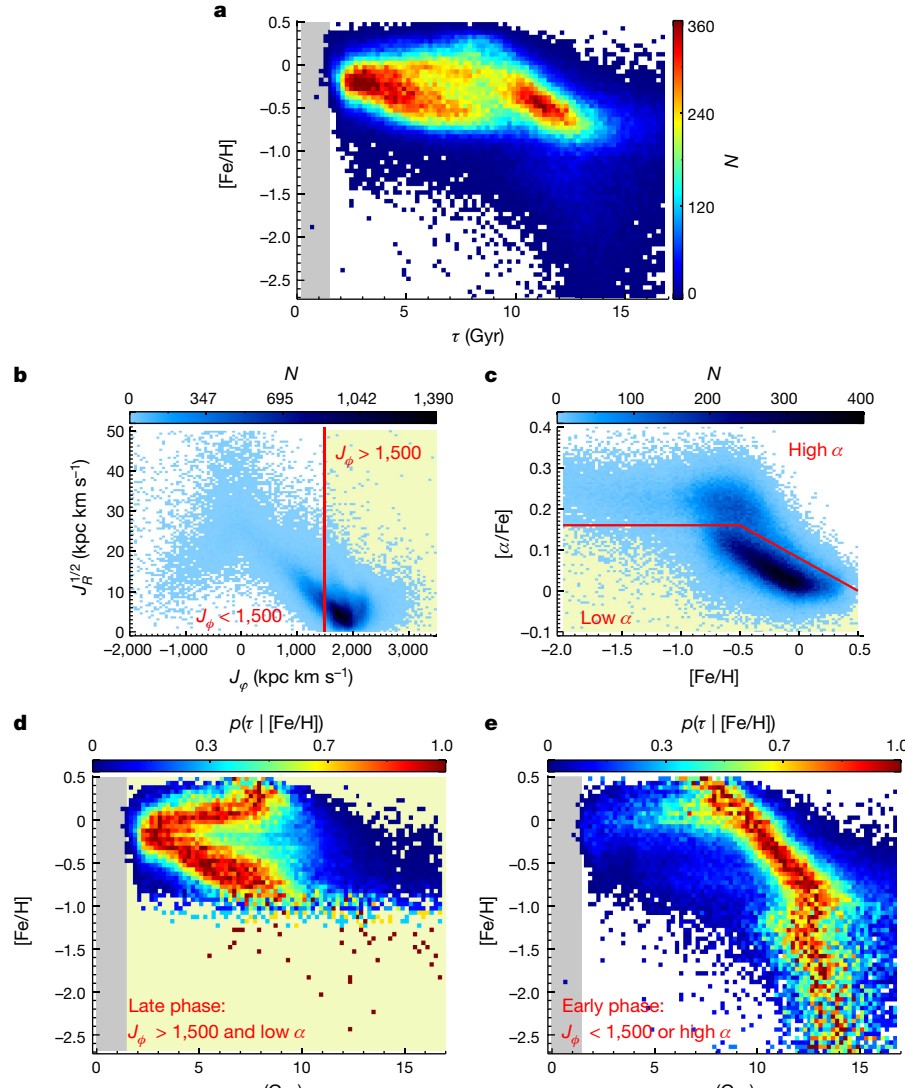

**Fig. 2 | Stellar age–metallicity relation revealed by our subgiant star sample. a**, Stellar distribution in the age–[Fe/H] plane for the whole subgiant star sample, colour-coded by the stellar number density, $N$. **b**, Stellar density distribution in the plane of the azimuthal action $J_\phi$ (equivalent to angular momentum $L_z$) versus radial action $J_R$. The vertical line delineates $J_\phi = 1,500$ kpc km s$^{-1}$, which separates the sample into high angular momentum (yellow background) and low angular momentum regimes. **c**, Stellar density distribution in the [Fe/H]–[$\alpha$/Fe] plane. The red solid line separates the sample into high-$\alpha$ and low-$\alpha$ (yellow background) regimes. **d**, Probability distribution of stellar age $p(\tau\,|\,[\text{Fe/H}])$, normalized to the peak value for each [Fe/H], for stars

with high angular momentum and low [$\alpha$/Fe] (yellow background regimes in **b** and **c**). **e**, Similar to **d** but for stars with low angular momentum or high [$\alpha$/Fe]. The two regimes exhibit a sharp distinction at $\tau \simeq 8$ Gyr. Prominent structures are shown for both regimes, such as the V-shaped structure in the late phase (**d**), and the metal-poor ([Fe/H] $\lesssim -1$) 'halo' and metal-rich ([Fe/H] $\gtrsim -1$) 'disk' sequences in the early phase (**e**). In the early phase, the two sequences merge at [Fe/H] $\simeq -1$, but the metal-rich sequence is older than the metal-poor sequence by around 2 Gyr at this metallicity, leading to a Z-shaped structure in $p(\tau\,|\,[\text{Fe/H}])$.

interval (Extended Data Fig. 1). Given the sequence's slope, this implies that the [Fe/H] dispersion at a given age is smaller than 0.22 dex across the 1.5 dex range in [Fe/H].

Both the halo and old disk sequences extend to [Fe/H] $\simeq -1$. However, at that [Fe/H] value, the old disk sequence is approximately 2 Gyr older than the halo sequence, leading to a Z-shaped structure in $p(\tau|[\text{Fe/H}])_{\text{early}}$. This feature is a second aspect of the distribution that has not, to our knowledge, been seen before[21].

### Formation and enrichment of the Milky Way's old disk

Tentative hints for some of these features in $p(\tau|[\text{Fe/H}])$ have been seen in earlier work[24,25] (see the discussion in the Supplementary Information) but these studies lacked the sample size or precision for definitive inferences about the Galactic formation history. Figure 2 shows clearly that the old,

high-$\alpha$ 'thick' disk of our Milky Way started to form approximately 13 Gyr ago, which is only 0.8 Gyr after the Big Bang[19], and extended over 5–6 Gyr, and the interstellar stellar medium (ISM) forming the stars was continually enriched by more than 1 dex, from [Fe/H] $\simeq -1$ to 0.5. The tightness of this [Fe/H]–age sequence implies that the ISM must have remained spatially mixed thoroughly during this entire period. Had there been any radial (or azimuthal) [Fe/H] variations (or gradients) in excess of 0.2 dex in the star-forming ISM at any time, this would have increased the resulting [Fe/H]–age scatter beyond what is seen. Such gradients, along with orbital migration, are the main reason that the later Galactic disk shows a considerably higher [Fe/H] dispersion at a given age[4,26]. The results also show that the formation of the Milky Way's old, $\alpha$-enhanced disk overlapped in time with the formation of the halo stars: the earliest disk stars are 1–2 Gyr older than the major halo populations at [Fe/H] $\simeq -1$ (see the Z-shaped structure).

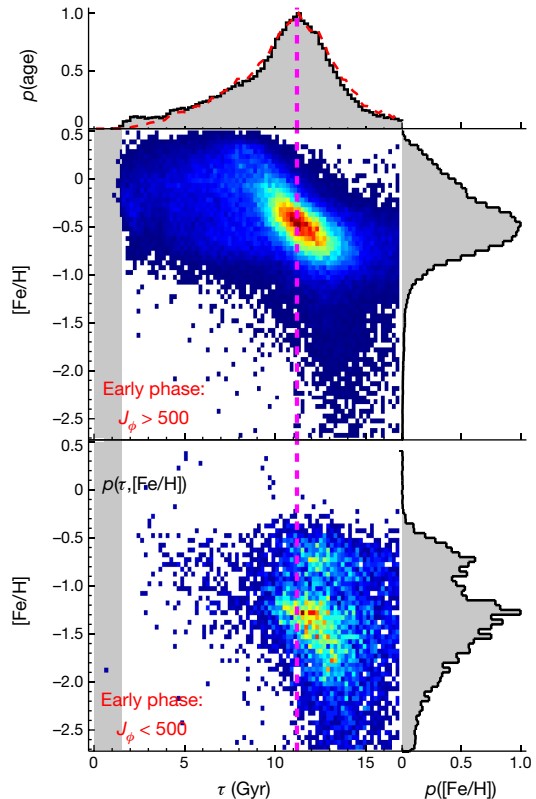

**Fig. 3 | Probability of stellar distribution in the $J_\phi$ versus [Fe/H] plane, $p(\tau, \text{[Fe/H]})$, for stars formed in the early phase.** The stars formed in the early phase are divided into $J_\phi > 500$ kpc km s$^{-1}$ (upper) and $J_\phi < 500$ kpc km s$^{-1}$ (lower). The stellar distribution probability is normalized to the peak value so that the colour from blue to red represents a value from 0 to unity. Note that this is different from $p(\tau\,|\,\text{[Fe/H]})$ in Fig. 2, which is normalized for each [Fe/H]. The histograms show the distribution integrated over [Fe/H] (top panel) or age (right panels). In the top panel, the age distribution $p(\tau)$ is a measure of the relative star-formation history. The dashed curve in red is the result after correcting for the volume selection effect. The vertical dashed line delineates a constant age of 11.2 Gyr, when the star-formation rate reaches its maximum.

In Fig. 3 we examine the $p(\tau\,|\,\text{[Fe/H]})_{\text{early}}$ distribution more closely by separating stars with at least modest angular momentum, $J_\phi > 500$ kpc km s$^{-1}$, from those stars on nearly radial or even retrograde orbits, $J_\phi < 500$ kpc km s$^{-1}$. This further sample differentiation by angular momentum leads again to two nearly disjoint $p(\tau\,|\,\text{[Fe/H]})$ distributions. The first (Fig. 3, upper panel), with mostly [Fe/H] > −1, is dominated by the tight $p(\tau\,|\,\text{[Fe/H]})$ sequence that we we have already attributed to the old disk. The second, predominately [Fe/H] < −1.2, reflects the halo.

Note that Fig. 3, lower panel shows a distinct set of stars with $J_\phi < 500$ kpc km s$^{-1}$, for which the $p(\tau\,|\,\text{[Fe/H]})$ locus indicates that they are the oldest and most metal-poor part of the old disk sequence (see also Extended Data Fig. 2). These stars indicate that some of the oldest members of the old disk sequence were present during an early merger event, by which they were 'splashed' to low-angular-momentum orbits[27,28]. This ancient merger event is presumably the merger with the Gaia-Enceladus satellite galaxy[11] (also known as Gaia Sausage[10]; hereafter Gaia-Sausage-Enceladus), which has contributed most of the Milky Way's halo stars[7,29]. The fact that the splashed old disk stars with very little angular momentum are exclusively seen at $\tau \gtrsim 11$ Gyr constitutes strong evidence that the major merger process between the old disk and the Gaia-Sausage-Enceladus satellite galaxy was largely completed 11 Gyr ago. This epoch is 1 Gyr earlier than previous estimates that were based on the lower age limit of the halo stars, 10 Gyr (refs. [11,21,30]).

Figure 3 shows the volume-corrected two-dimensional distribution $p(\tau, \text{[Fe/H]})$ (see the Supplementary Information for the correction of the volume selection effect), rather than the $p(\tau\,|\,\text{[Fe/H]})$ of Fig. 2. Figure 3 reveals a remarkable feature, namely that the star-formation rate of the old disk reached a prominent maximum at around 11.2 Gyr ago, apparently just when the merger with the Gaia-Sausage-Enceladus satellite galaxy was completed, and then continuously declined with time. The most obvious interpretation of this coincidence is that the perturbation from the Gaia-Sausage-Enceladus satellite galaxy greatly enhanced the star formation of the old disk. Note that this star-formation peak among the old disk stars ~11 Gyr ago is very consistent with earlier indications of such a peak based on abundances only[31].

To put our results into the bigger picture of galaxy formation and evolution, the multiple assembly phases are seen to be universal among present-day star-forming galaxies. Using the IllustriesTNG simulation, Wang et al.[32] showed that galaxy mergers and interactions have played a crucial role in inducing gas inflow, resulting in multiple star formation episodes, intermitted by quiescent phases. Observationally, the best testbed for this theoretical picture would be here at home within our Galaxy. Our study has demonstrated the power of such tests for galactic assembly and enrichment history in the full cosmic timeline, from the very early epoch ($\tau \simeq 13$ Gyr or redshift $z > 10$) to the current time.

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

## Methods

### Stellar labels from spectroscopy

Building this sample of subgiant stars with precise ages, abundances and orbits requires a number of steps. The first step is to derive stellar atmospheric parameters from the LAMOST DR7 spectra, which we did using the data-driven Payne (DD-Payne) approach, verified in detail using analogous data from LAMOST DR5 (ref. [33]). This leads to a catalogue of effective temperature $T_{eff}$, surface gravity $\log g$, microturbulent velocity $v_{mic}$ and elemental abundance for 16 elements (C, N, O, Na, Mg, Al, Si, Ca, Ti, Cr, Mn, Fe, Co, Ni, Cu, Ba) values for 7 million stars. We also derive an $\alpha$-element to iron abundance ratio [$\alpha$/Fe], which will serve in the age estimation to identify the right set of isochrones for each object. For a spectral signal-to-noise ratio ($S/N$) higher than 50, the typical measurement uncertainties are about 30 K in $T_{eff}$ and 0.05 dex in the abundances we use here: [Fe/H] and [$\alpha$/Fe] (ref. [33]).

### Absolute magnitude and spectroscopic parallax

Determining accurate and precise absolute magnitudes is crucial for age determination of subgiant stars (Fig. 1a). The Gaia astrometry provides high-precision parallax for stars within approximately 2 kpc, whereas for more distant stars the Gaia parallaxes have uncertainties in excess of 10%. For these distant stars, spectroscopic estimates of absolute magnitude are needed to ensure precise age determination. We derive $M_K$, the absolute magnitude in the Two Micron All Sky Survey (2MASS) K band, from the LAMOST spectra, using a data-driven method based on neural network modelling (see Supplementary Information for details). Extended Data Figure 3 illustrates that for LAMOST spectra with high signal-to-noise ratio ($S/N > 80$), our spectroscopic $M_K$ estimates are precise to better than 0.1 mag at [Fe/H] = 0 (and 0.15 mag at [Fe/H] = −1). Furthermore, a comparison between spectroscopic $M_K$ and geometric $M_K$ from Gaia parallaxes provides an efficient way of identifying unresolved binaries[33,34] (Extended Data Fig. 3). For the subsequent modelling, we combine these two approaches through a weighted mean algorithm

$$M_K = \frac{M_K^{\text{geom}}/\sigma_{\text{geom}}^2 + M_K^{\text{spec}}/\sigma_{\text{spec}}^2}{\sigma_{\text{spec}}^{-2} + \sigma_{\text{geom}}^{-2}}. \tag{2}$$

Here $M_K^{\text{geom}}$ refers to the geometric $M_K$, i.e., $M_K$ derived using Gaia parallax, $M_K^{\text{spec}}$ the spectroscopic $M_K$ estimates, and $\sigma$ the uncertainty in the $M_K$ estimates. We are then in a position to select subgiant stars as lying between the two straight lines in the $T_{eff}$–$M_K$ diagram. As isochrones depend on [Fe/H], this is done separately for each [Fe/H] bin, with the adopted slopes and intercepts for the boundary lines presented in Extended Data Table 1. As an example, the boundaries for stars with solar metallicity are shown in the Fig. 1a. To ensure the boundaries vary smoothly with [Fe/H], we interpolate the slopes and intercepts listed in Extended Data Table 1 to match the measured [Fe/H] for each star.

### Cleaning sample cuts

To have a subgiant star sample with high purity, we have applied cleaning criteria to discard stars with poor data quality or stars that are possible contaminations of the subgiant sample.

- We discard unresolved binaries that we identify through differences in their spectro-photometric parallax and their geometric parallax from Gaia, by requiring

$$\frac{\varpi_{\text{spec-photo}} - \varpi_{\text{geom}}}{\sqrt{\sigma_{\text{spec}}^2 + \sigma_{\text{geom}}^2}} > 2 \tag{3}$$

Here $\varpi_{\text{spec-photo}}$ is the spectro-photometric parallax deduced from the distance modulus using the spectroscopic $M_K$ and 2MASS apparent magnitudes[35].

- We discard stars with spurious Gaia astrometry by requiring a Gaia re-normalized unit weight error (RUWE) larger than 1.2 or an astrometric fidelity less than 0.8 (ref. [36]).
- We discard stars that show significant flux variability according to the variation amplitude of the Gaia magnitudes between different epochs,

$$\Delta_G = \frac{\sqrt{\text{PHOT\_G\_N\_OBS}}}{\text{PHOT\_G\_MEAN\_FLUX\_OVER\_ERROR}} \tag{4}$$

where PHOT_G_N_OBS is the number of epochs, and PHOT_G_MEAN_FLUX_OVER_ERROR is the mean flux over error ratio for Gaia G-band photometry. We calculate the ensemble median ($\overline{\Delta_G}$) and dispersion $\sigma(\Delta_G)$ of $\Delta_G$ as a function of G-band magnitude and define any one star as a variable if

$$\frac{\Delta_G - \overline{\Delta_G}}{\sigma(\Delta_G)} > 3 \tag{5}$$

Most of the variables eliminated by this criterion are found to be pre-main-sequence stars.

- We discard stars that are less luminous than the subgiant branch of a 20 Gyr isochrone, which is the boundary of our isochrone grid. Such stars are mainly contaminations of either pre-main-sequence stars or main-sequence binary stars that survived elimination by the above criteria.
- We discard all stars with $M_K$ brighter than 0.5 mag to avoid contamination from He-burning horizontal branch stars. This comes at a price: we eliminate essentially all stars younger than about 1.5 Gyr.
- We require all stars in our sample to have LAMOST spectral $S/N > 20$ and to have good DD-Payne fits, by requiring 'qflag_$\chi^2$ = good'[33]. We further restrict our stars to have $T_{eff} < 6,800$ K, where DD-Payne abundances are most robust.

After these cleaning cuts, the remaining sample contains 247,104 stars (Fig. 1), all of which are presumed to be subgiants.

### Age estimates by isochrones

The ages of the subgiant sample stars are determined by matching the Gaia astrometric parallax $\varpi$, the LAMOST spectroscopic stellar parameters $T_{eff}$, $M_K$, [Fe/H] and [$\alpha$/Fe], and the Gaia and 2MASS photometry in the G, BP, RP, J, H and K bands with the YY stellar isochrones[18,37] using a Bayesian approach (see Supplementary Information for details) . Note that in our Bayesian model we have chosen not to impose a prior that all stars should be younger than the current knowledge of the age of the Universe from the cosmic microwave background measurements of Planck (13.8 Gyr)[19]. This is for two main reasons. First, the upper limit of the stellar age is an independent examination of the age of the Universe, whereas imposing age priors on the inference from the cosmological model might induce bias into the results. Second, imposing an upper age limit may increase the complexity of the statistics.

To convert the Gaia parallax to absolute magnitudes, we also need to know the extinction. Therefore, we have determined the reddening and extinction for individual stars using intrinsic colours empirically inferred from their stellar parameters (see Supplementary Information for details).

We have also tested the age estimation using other public isochrones, such as the MIST[38,39], and find that, in the case of the solar $\alpha$-mixture, the age estimates based on YY and MIST show good consistency except for the fact that the MST isochrones predict ages older by 0.5 Gyr (Extended Data Fig. 4). However, the $\alpha$-element enhancement, which is not available in the current public MIST isochrones, has a large impact on the age estimation, and ignoring the $\alpha$-element enhancement will lead to an overestimate of stellar age by up to 2 Gyr for old stars (Extended Data Fig. 4). Ages from the YY isochrones seem to be reasonable as the ages of the oldest stars are comparable to the age of the Universe (Fig. 2).

## Orbital actions

Using the radial velocity from the LAMOST data, proper motions from Gaia and a combination of spectro-photometric distance and geometric distance (see Supplementary Information for details), we compute the orbital actions $(J_R, J_\phi, J_Z)$ and the angles of our sample stars using galpy[40], assuming the MWPotential2014 potential model. We assume that the Sun is located at $R_\odot = 8.178$ kpc (ref. [41]) and $Z_\odot = 10$ pc above the disk mid-plane[42]. We assume the local standard of rest LSR = 220 km s$^{-1}$, and the solar motion with respect to the LSR to be $(U_\odot, V_\odot, W_\odot) = (-7.01$ km s$^{-1}$, 10.13 km s$^{-1}$, 4.95 km s$^{-1})$ (ref. [43]).

## Accounting for selection effects

To verify that our findings are not caused by artefacts due to selection effects, we adopt two approaches to address this issue. First, we apply our target selection to the Gaia mock catalogue of Rybizki et al.[44] and investigate the age–[Fe/H] relation (Extended Data Fig. 5). Second, we directly correct for the volume selection function of our sample to account for the fact that, for a given line of sight, older subgiant stars probe to a smaller distance than the younger stars as the former are fainter. The age distribution of the thick disk stars after applying the selection function correction is illustrated in Fig. 3. Eventually, we concluded that the selection function has a negligible impact on our conclusions (see Supplementary Information for more details).

In addition, we have compared the stellar age–[Fe/H] relation from our sample with literature results for both stars[25] and globular clusters[45–47] that have robust age estimates (Extended Data Fig. 6). The comparisons are qualitatively consistent, albeit the literature samples are too small to draw a clear picture of the assembly and enrichment history of our Galaxy (see Supplementary Information for a detailed discussion).

## Data availability

The Gaia eDR3 data is public available at https://www.cosmos.esa.int/web/gaia/earlydr3 The LAMOST DR7 spectra data set is public available at http://dr7.lamost.org. The subgiant star catalogue generated and analysed in this study is provided as Supplementary Table 1, and it can also be reached through a temporary path https://keeper.mpdl.mpg.de/d/019ec71212934847bfed/. The YY isochrones adopted for age determination in this work is public available at http://www.astro.yale.edu/demarque/yyiso.html.

## Code availability

The stellar orbit computation tool galpy adopted in this work is public available at http://github.com/jobovy/galpy. The DD-Payne code adopted for determining stellar labels, the neural network code for determining $M_K$ from the LAMOST spectra and the Bayesian code for stellar age estimation are currently not publicly accessible online, as they are a part of ongoing survey data analysis efforts that will be applied to the upcoming LAMOST survey spectrum set. However, the codes can be shared on request.

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

**Acknowledgements** We thank D. Xu and N. Frankel for helpful discussion, and J. Rybizki for his kind help with using the Gaia mock catalogues. M.X. acknowledges partial support from NSFC grant no. 11833006 for his academic visit to NAOC from November 2021 to January 2022. This work has used data products from the Guoshoujing Telescope (LAMOST). LAMOST is a National Major Scientific Project built by the Chinese Academy of Sciences. Funding for the project has been provided by the National Development and Reform Commission. LAMOST is operated and managed by the National Astronomical Observatories, Chinese Academy of Sciences. This work has made use of data products from the European Space Agency (ESA) space mission Gaia. Gaia data are being processed by the Gaia Data Processing and Analysis Consortium (DPAC). Funding for the DPAC is provided by national institutions, in particular the institutions participating in the Gaia MultiLateral Agreement. The Gaia mission website is https://www.cosmos.esa.int/gaia. The Gaia archive website is https://archives.esac.esa.int/gaia. This publication has also used data products from the 2MASS, which is a joint project of the University of Massachusetts and the Infrared Processing and Analysis Center/California Institute of Technology, funded by the National Aeronautics and Space Administration and the National Science Foundation.

**Author contributions** M.X. conducted the construction of the subgiant sample and the determination of stellar parameters and ages. M.X. and H.-W.R. jointly executed the data analysis and manuscript writing.

**Funding** Open access funding provided by Max Planck Society.

**Competing interests** The authors declare no competing interests.

**Additional information**
**Correspondence and requests for materials** should be addressed to Maosheng Xiang or Hans-Walter Rix.

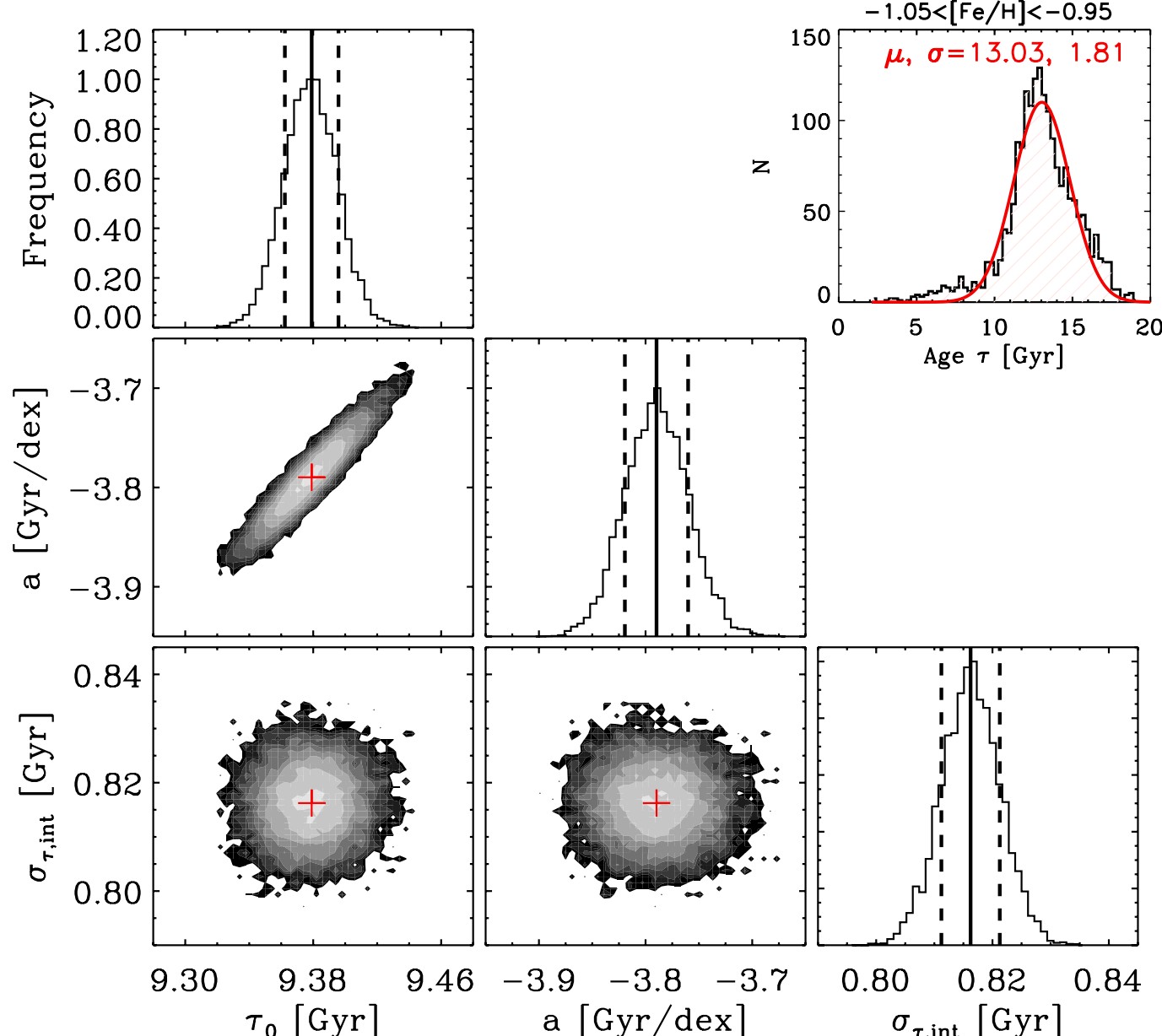

**Extended Data Fig. 1 | MCMC determination of the intrinsic scatter of the age distribution of old, high-α ('thick') disk sequence, $P(\tau|[Fe/H])$, shown in panel (e) of Fig. 2.** The parameters shown are: $\sigma_{\tau,int}$ – the intrinsic age scatter, $\bar{\tau}_0$ – the mean stellar age at solar metallicity ([Fe/H] = 0), and a – the slope of mean age as a function of [Fe/H]. Specifically, we assume the age distribution for given [Fe/H] is $P(\tau, \delta\tau|[Fe/H], \bar{\tau}_0, a, \sigma_{\tau,int}) = G\left(\tau - \bar{\tau}([Fe/H]), \sqrt{\sigma_{\tau,int}^2 + \delta\tau^2}\right)$, where $G$ is the Gaussian function, $\delta\tau$ the measurement error of the age $\tau$, and $\bar{\tau}([Fe/H]) = \bar{\tau}_0 + a \times [Fe/H]$ (see Supplementary Information for details). Vertical solid and dashed lines indicate the mean and 1σ values of the estimated parameters. The resultant upper limit of the intrinsic age scatter $\sigma_{\tau,int}$ of the 'thick' disk sequence is -0.82 ± 0.01 Gyr. This indicates that, at a constant age, the upper limit of the 'thick' disk intrinsic [Fe/H] dispersion is 0.22 dex. The upper-right corner shows the age distribution for stars formed in the early phase but with −1.05 < [Fe/H] < −0.95, $J_\phi$ > 500 kpc.km/s – presumably the oldest thick disk stars. A Gaussian fit to the distribution (red curve) yields a mean age of 13 Gyr.

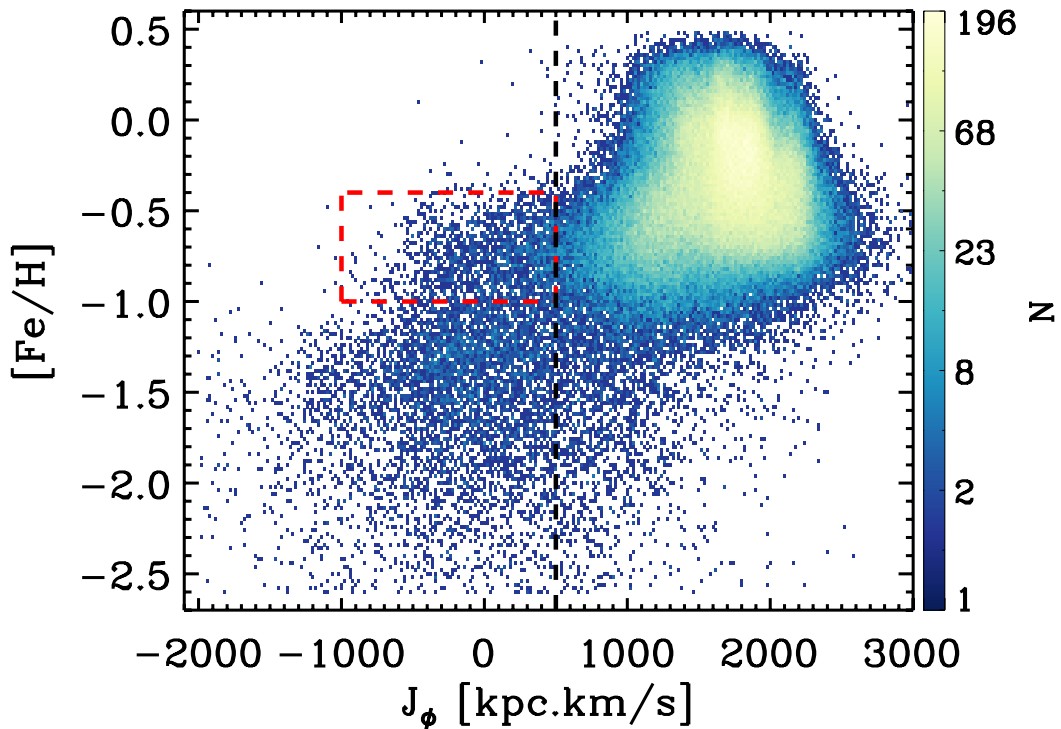

**Extended Data Fig. 2 | Stellar density distribution in the $J_\phi$ versus [Fe/H] plane.** The vertical line delineates a constant $J_\phi$ of 500 kpc.km/s, which we adopt to separate the kinematic halo from the kinematic 'thick' disk in Fig. 3. There is a tail of low-angular-momentum stars ($J_\phi < 500$ kpc.km/s) in the metallicity range of $-1 \lesssim$ [Fe/H] $\lesssim -0.4$ (box delineated by red dashed lines), presumably the 'splashed' thick disk stars due to the merger with the Gaia-Sausage-Enceladus satellite galaxy.

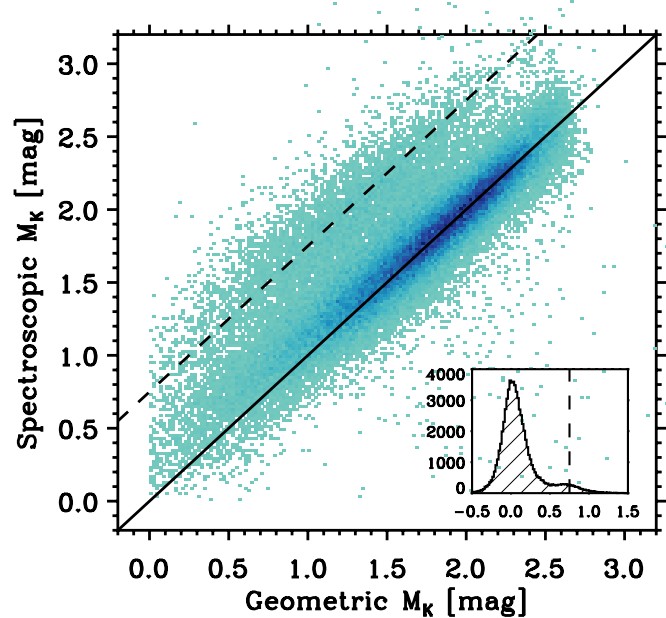

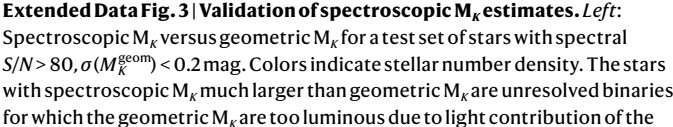

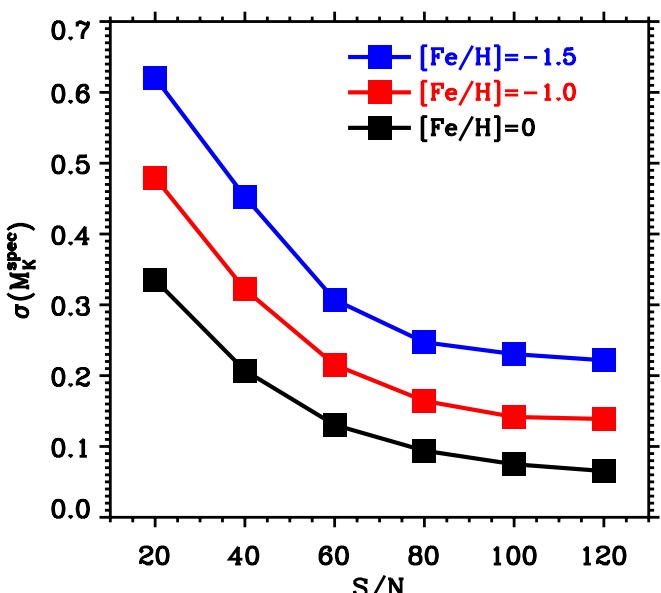

**Extended Data Fig. 3 | Validation of spectroscopic $M_K$ estimates.** *Left*: Spectroscopic $M_K$ versus geometric $M_K$ for a test set of stars with spectral $S/N > 80$, $\sigma(M_K^{geom}) < 0.2$ mag. Colors indicate stellar number density. The stars with spectroscopic $M_K$ much larger than geometric $M_K$ are unresolved binaries, for which the geometric $M_K$ are too luminous due to light contribution of the secondary. The solid line indicates the 1:1 line, and the dashed line indicates an offset of 0.75 mag, which corresponds to the case of equal-mass binaries. The small window in the panel shows a histogram of the difference for spectroscopic $M_K$ minus geometric $M_K$. *Right*: uncertainty of the spectroscopic $M_K$ estimates as a function of S/N, for subgiant stars of different metallicities.

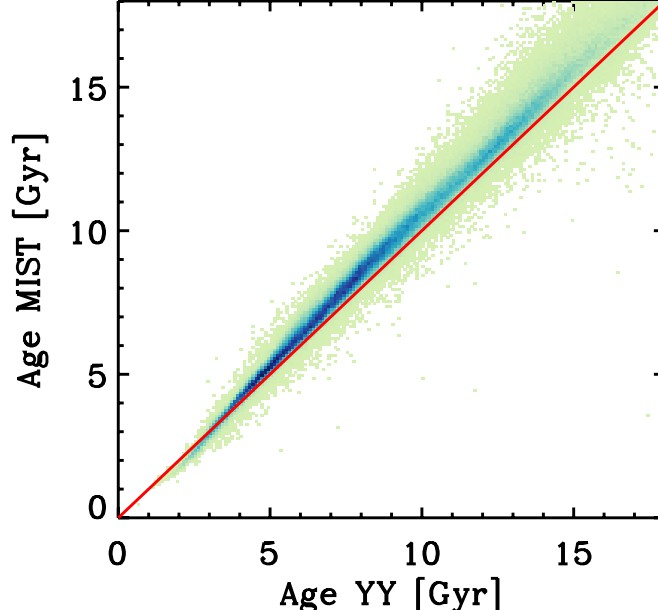

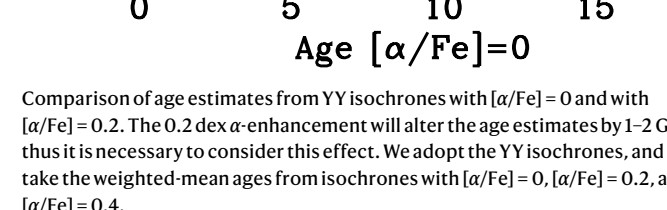

**Extended Data Fig. 4 | Illustration of age estimates from different isochrones.** *Left*: comparison of age estimates from YY (X-axis) and MIST iscohrones (Y-axis), both with [$\alpha$/Fe] = 0. MIST isochrones yield about 0.5 Gyr older ages. Currently, MIST isochrones are publically available only with [$\alpha$/Fe] = 0, while YY isochrones with different [$\alpha$/Fe] are available. *Right*: Comparison of age estimates from YY isochrones with [$\alpha$/Fe] = 0 and with [$\alpha$/Fe] = 0.2. The 0.2 dex $\alpha$-enhancement will alter the age estimates by 1–2 Gyr, thus it is necessary to consider this effect. We adopt the YY isochrones, and take the weighted-mean ages from isochrones with [$\alpha$/Fe] = 0, [$\alpha$/Fe] = 0.2, and [$\alpha$/Fe] = 0.4.

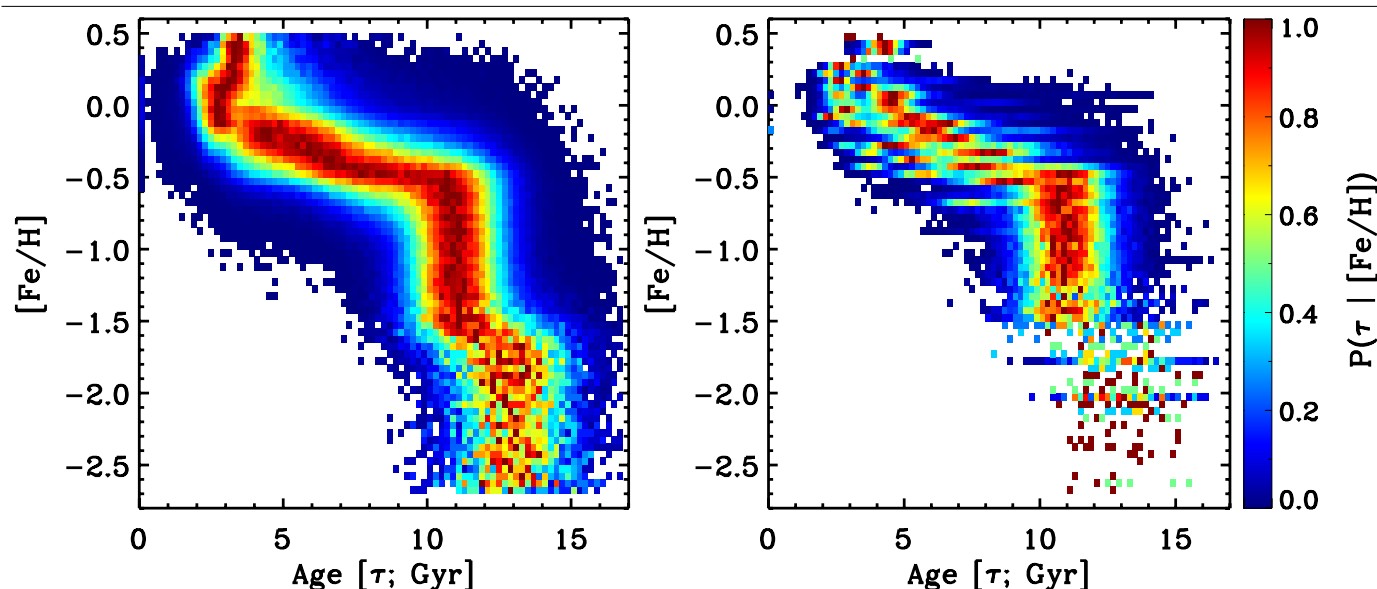

**Extended Data Fig. 5 | Examination of selection effect through Gaia Mock data.** *Left panel*: Age – [Fe/H] relation for subgiant stars in the Gaia mock catalog of Rybizki et al. [44]. The sample includes about 1,250,000 subgiant stars that in the same footprint and magnitude ranges as for the LAMOST. *Right panel*: Same as the *left* panel, but for a subset of the Gaia mock subgiant stars that has comparable number of the LAMOST sample (about 250,000 stars) randomly drawn from the sample shown in the *left* panel. Compared to the *left* panel, there are some artifacts for the younger populations ($\tau < 9$ Gyr) due to the smaller sample size, but this will not change the conclusion.

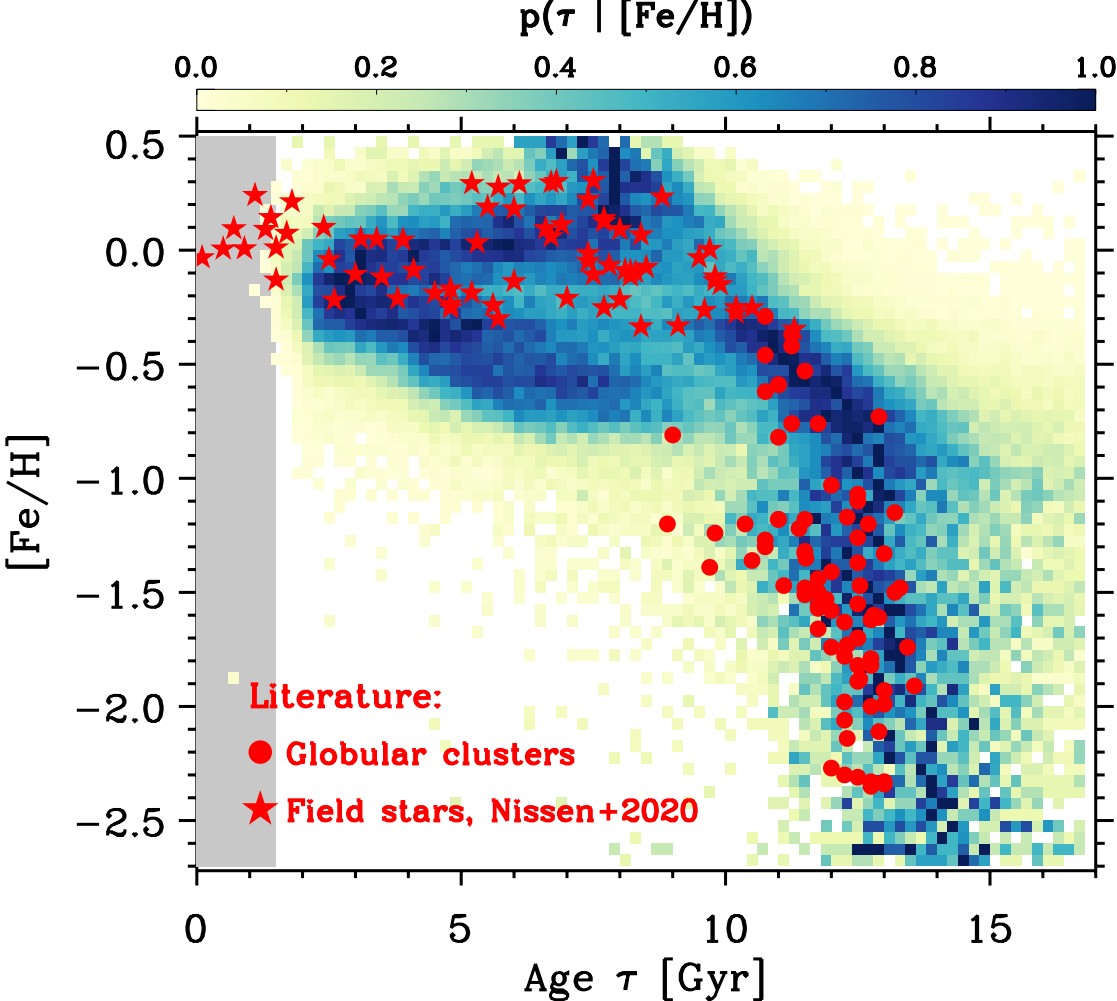

**Extended Data Fig. 6 | Comparison of the age-metallicity relation with literature.** The five-point stars in red represent field stars from Nissen et al.[25], while the dots in red are globular clusters (GCs) compiled from Forbes et al.[45], VandenBerg et al.[46], and Cohen et al.[47].

**Extended Data Table 1 | Slope and intercept of the linear functions for the upper and lower boundary of the subgiant star sample selection**

| [Fe/H] | slope$_1$ | zpt$_1$ | slope$_2$ | zpt$_2$ |
|---|---|---|---|---|
| 0.4 | −0.005 | 26.00 | −0.0014 | 10.00 |
| 0.2 | −0.005 | 26.25 | −0.0014 | 10.10 |
| 0.0 | −0.005 | 26.50 | −0.0014 | 10.20 |
| −0.2 | −0.0045 | 24.25 | −0.0014 | 10.30 |
| −0.4 | −0.004 | 22.00 | −0.0014 | 10.50 |
| −0.6 | −0.004 | 22.20 | −0.0014 | 10.70 |
| −0.8 | −0.004 | 22.60 | −0.00125 | 9.90 |
| −1.0 | −0.004 | 23.00 | −0.00125 | 10.00 |
| −1.2 | −0.004 | 23.20 | −0.001 | 8.65 |
| −1.4 | −0.004 | 23.40 | −0.001 | 8.70 |
| −1.6 | −0.004 | 23.60 | −0.001 | 8.75 |
| −1.8 | −0.004 | 23.80 | −0.001 | 8.80 |
| −2.0 | −0.004 | 24.00 | −0.001 | 8.85 |
| −2.2 | −0.004 | 24.20 | −0.001 | 8.90 |
| −2.5 | −0.004 | 24.20 | −0.001 | 8.95 |

The boundary of subgiant stars in the $T_{eff}$ – $M_K$ diagram is $M_K$ = slope × $T_{eff}$ + zpt. The slopes and intercepts ('zpt') listed in the table are adopted as anchors for interpolation to match the measured [Fe/H] of each star.