## [Peer Review File · Nature]

Manuscript Title: 2021-11-17398A

Reviewer Comments & Author Rebuttals

Reviewer Reports on the Initial Version:

Referee #1 (Remarks to the Author):

This paper reports on a powerful new probe of the formation history of the Milky Way, based on ages and a limited set of abundances for a significant sample of stars. This is, in fact, the first such paper to my knowledge to be able to draw such a clear picture of this history.

As such, it is entirely appropriate for publication in Nature.

The arguments are well described, in general, and convincing in my view.

The detailed Methods section addresses most all of the questions I had about the quantities used in this analysis. The statistical tools employed are appropriate.

I have a few additional comments which I hope the authors will take into account in their revisions.

1) I believe the use of "assemblage" as a verb in the title is incorrect. Rather, the verb "assembly" is preferable.

2) Although the paper is generally very well written, there are numerous places with articles such as "the", "an", etc. are omitted, or where the word is close, but not quite correct -- e.g., "lead" when "led" is meant, and others. Presumably these would be caught in the editing

process, but they should be addressed in the revision -- easiest way would be to have a native English speaker go through the text with a fine-tooth comb.

3) There is inconsistent use of multiple modifiers. Correctly, as in the 2nd line of the abstract -- "star-formation history" but not used at all in some cases throughout the rest of the manuscript, for example on page 5, in the partial paragraph following eqn (1) -- "element enrichment history"  "element-enrichment history".

4) The use of subgiant stars as age probes is entirely appropriate, but the authors should make it clear just how brief this stage of evolution truly is. In fact, the short time interval during which a star finds itself in this evolutionary stage makes this a **more powerful** age probe, hence worthy of emphasizing. This is mentioned in passing, but I believe it should be underscored, in particular for the non-specialist.

5) Incorrect use of "dex" as a unit. In fact, "dex" is a scale, so its use is appropriate when referring to offsets or dispersions, but not in the sense it is throughout the manuscript -- e.g., "[Fe/H] = -1 dex"  "[Fe/H] = -1"

Author Rebuttals to Initial Comments:

Referee #1 (Remarks to the Author):

This paper reports on a powerful new probe of the formation history of the Milky Way, based on ages and a limited set of abundances for a significant sample of stars. This is, in fact, the first such paper to my knowledge to be able to draw such a clear picture of this history. As such, it is entirely appropriate for publication in Nature.

The arguments are well described, in general, and convincing in my view. The detailed Methods section addresses most all of the questions I had about the quantities used in this analysis. The statistical tools employed are appropriate.

I have a few additional comments which I hope the authors will take into account in their revisions.

We are grateful to the anonymous referee for these very positive and inspiring comments. We have revised the manuscript carefully following these comments. Below we present the reply to the comments accordingly.

1) I believe the use of "assemblage" as a verb in the title is incorrect. Rather, the verb "assembly" is preferable.

Reply: we have now adopted the verb "assembly" throughout the manuscript.

2) Although the paper is generally very well written, there are numerous places with articles such as "the", "an", etc. are omitted, or where the word is close, but not quite correct -- e.g., "lead" when "led" is meant, and others. Presumably these would be caught in the editing process, but they should be addressed in the revision -- easiest way would be to have a native English speaker go through the text with a fine-tooth comb.

Reply: we have carefully read the manuscript to correct for typos, and we have used the software tool "Writefull" on overleaf to help in this editing. We hope the Nature language editors will help double check for any possible remaining typos.

3) There is inconsistent use of multiple modifiers. Correctly, as in the 2nd line of the abstract -- "star-formation history" but not used at all in some cases throughout the rest of the manuscript, for example on page 5, in the partial paragraph following eqn (1) -- "element enrichment history"  "element-enrichment history".

Reply: thank you for this correction. We have now unified the format.

4) The use of subgiant stars as age probes is entirely appropriate, but the authors should make it clear just how brief this stage of evolution truly is. In fact, the short time interval during which a star finds itself in this evolutionary stage makes this a *more powerful* age probe, hence worthy of emphasizing. This is mentioned in passing, but I believe it should be underscored, in particular for the non-specialist.

Reply: we have rephrased the text for emphasizing this point.

5) Incorrect use of "dex" as a unit. In fact, "dex" is a scale, so its use is appropriate when referring to offsets or dispersions, but not in the sense it is throughout the manuscript -- e.g., "[Fe/H] = -1 dex"  "[Fe/H] = -1"

Reply: thank you for pointing this out. We have revised the text accordingly.